# Electrodeposition from a Graphene Bath: A Sustainable Copper Composite Alloy in a Graphene Matrix

**Hayley Richardson, Charles Bopp, Bao Ha, Reeba Thomas and Kalathur S.V. Santhanam *** 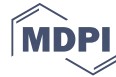

School of Chemistry and Materials Science, Rochester Institute of Technology, Rochester, NY 14623, USA; hrk3290@rit.edu (H.R.); cbs9344@rit.edu (C.B.); bha9120@rit.edu (B.H.); rt5228@rit.edu (R.T.)
* Correspondence: ksssch@rit.edu

**Abstract:** The leaching effect of metals has led to the introduction of government regulations for the safety of the environment and humans. This has led to the search for new alloys with long-lasting sustainability. Herein, we wish to report a new brass alloy containing carbon with a remarkable sustainability produced by electrodeposition from a graphene quantum dots bath. The electrochemical measurements were carried out using cyclic voltammetry, potentiodynamic analysis, and Tafel measurements, and the deposits were characterized by X-ray fluorescence spectroscopy (XRF), Raman imaging, X-ray diffraction (XRD), atomic force microscopy (AFM), and scanning electron microscopy (SEM) to understand the surface morphology and elemental compositions. The current–time transients in the potential-step electrolysis were used to investigate the nucleation and growth mechanism. The smooth and compact deposit obtained at $-0.60$ V showed a composition of Cu = 24.33 wt %; Zn = 0.089 wt %; and C = 75.57 wt %. The SEM and energy dispersion X-ray analysis revealed a surface morphology with a uniform distribution of the particles and the presence of Cu, Zn, and C. The corrosion density of the material is very much lower than that of conventional brass, suggesting a higher sustainability.

**Keywords:** copper composite with zinc; graphene quantum dots; cyclic voltammetry; Tafel analysis; XRF; SEM; nucleation modeling

## 1. Introduction

Brass (Cu-Zn alloys) is used for variety of purposes, such as in water management, musical instruments, and electronic devices. The general concern is how safe would be its long-term usage in potable water systems. Many countries, such as the USA, Canada, Japan, and those of the European Union, have started examining the consequences of using the brass value chain [1–3] based on its sustainability. The development of new materials should provide a safety for the future generations and should be accomplished economically. The alloy should be lead free as its ions have long-term toxic effects. The US Federal government introduced a bill of "Reduction of lead in drinking water" on 4 January 2011, which became an act from 4 January 2014 (RLDWA). The leaching effects of zinc from brass have been of a concern in several electronic devices and hence limiting the use of Cu (85%)-Zn (15%) needs to be considered. In relevance to the new Cu-Zn-C alloy, the concern of lead contamination is totally absent. In electronic usage, the zinc content in the alloy is extremely small (<<15%), thus providing economic stability.

Copper and zinc are essential metals available on earth with the earth's crust having 50 ppm copper (about $7.2 \times 10^8$ metric tons) and 75 ppm ($10.8 \times 10^8$ metric tons) zinc [4]. The alloying of these two metals results in a number of intermetallics with different ratios of the combination, collectively called brass, which is of historical importance since being electrochemically produced in 1841 [5]. Brass is highly malleable and workable. Several musical instruments, such as harmonicas, organ pipes, electric guitars, etc., require brass in their construction. White brass represents a class with a low copper and zinc content, giving

a silvery color. Electrochemically alpha brass with a copper content higher than 70 wt % has been deposited [6]. The electrochemical deposition requires the complexing agent in order to shift the electrochemical reduction potentials of the copper and zinc, conducive to the formation of the intermetallic. Copper forms a stronger complex than zinc with a complexing agent like cyanide, such that in this medium a co-deposition could be achieved. In a cyanide bath a low porosity brass has been produced. As the toxicity of cyanide is an environmental concern in the disposal of the waste solution, several other alternatives have been proposed. Other complexing agents, such as EDTA, sorbitol, tartrate, pyrophosphate, gluconate, and D-mannitol, have been successfully employed [7–14]. However, the deposits of brass in the non-toxic complexing agents do not reach the quality that is obtained in the cyanide medium due to oxides and hydroxides formation. In order to produce better deposits, mixed ligands have been employed [14,15]. The addition of allyl alcohol has been shown to improve the performance of the deposit.

A mixture of alpha and beta brass has been produced using ionic liquids such as 1-ethyl-3-methylimidazolium chloride [16]. Other ionic liquids, such as choline, produces nano deposits of brass [17–20]. A deep eutectic solvent has been used for the electrolytic deposition of Cu-Zn [21]. The potentiostatic deposition of brass in ionic liquid has resulted in a high (90 wt %) copper content with a mirror-bright reflectivity [22]; the mirror-bright reflectivity and adherent layers have been attributed to the increased concentration of zinc concentration in the bath. Survila et al. [11] have investigated the kinetics of partial processes and their correlation in zinc–copper co-deposition in a pH ranging from 4.5 to 10.

The nucleation and growth of metal atoms during the electrochemical processes have been investigated in several reports [23–32]. The 2D monolayer underpotential deposition of Cu has provided an advantage for the control of the Au nanoparticles [25] and the 2D-to-3D transition has been studied by potentiodynamic studies. The nucleation processes in the deposition of copper, zinc, and cobalt in graphene quantum dot baths have been discussed for understanding the mechanism [33–36].

The electrodeposition of metals has been of great importance in industrial applications. The electrodeposition of copper is used in microelectronics for interconnections [37–40]. The copper foils are used as current collectors in lithium batteries and the 3D scaffolds have interface energy with Li. By atomically depositing zinc, the interfacial energy has been reduced [41]. A dendrite-free Li metal could be deposited on a $Cu_{99}$-Zn alloy [42]. The alloy has been shown to be effective in the electrochemical reduction of $CO_2$ to ethylene, which is attributed to the proximity of the Cu and Zn atoms [41], causing an enhanced reduction of carbon dioxide to CO and HCOOH [42,43]. The electrocatalytic activity of the alloy in hydrogen production has been shown to be due to the development of new phases upon polarization [44]. The alloy placed on a screen-printed electrode has been shown to be useful for enzyme-free superoxide sensing [45]. Alloy films have been investigated for a shape-memory alloy [46]; the electrodeposition of thin films provides a higher power-to-volume ratio than the bulk shape-memory alloys due to microstructural differences in the two methods. The shape-memory alloy has been tested in bending applications.

The Cu-Zn alloy electrodeposition requires rigorous bath conditions, such as a complexing agent, a potential, or current control, which results in different microstructures. Prior to the environmental regulations imposed by the governments, a cyanide bath had been very popular as it produced defect-free deposits. As a consequence, several investigations have been carried out in non-cyanide baths, such as citrate [47–50], ethylene diamine tetra acetic acid (EDTA) [8,51], glycerol [52,53], triethanolamine [54], glucoheptonate [55], nitroloacetic acid [56], tartarate [22], choline acetate [57], and bis(trifluormethylsulfonyl) imide [58]. The relative merits of the different baths have been discussed in the literature [53,59]. In the literature, alloys with high Cu (>65 wt %) and Zn (35 wt %) have been reported [60]. The nucleation and growth mechanism in the formation of the alloy has not been well understood. In this paper, we wish to report on an alloy with a low Zn and high Cu content along with carbon inserted into the alloy by using graphene quantum dots. This new brass has a remarkable sustainability in comparison to normal brass. Its low corrosion

rate prevents dezincification, which is desirable for many technological applications. The nucleation and growth mechanism during the deposition was also investigated during the formation of these three component systems. This is probably the first report of a carbon-containing brass, opening up several applications.

## 2. Experimental

Chemicals: Cupric sulfate ($CuSO_4.5H_2O$) (ACS grade, Batch no. 0000106873) was obtained from MACRON (Avantor, Pa, Allentown, PA, USA). Zinc sulfate ($ZnSO_4.7H_2O$) (99%; lot SLBK8996V) and potassium sulfate ($K_2SO_4$) were obtained from Sigma Aldrich (St. Louis, MO, USA). A graphene solution (1% *w/v*) was made using a TTK-4 graphite plate (Ohio carbon Black) using the procedure described earlier [61,62]. Graphite electrodes were kept in intimate contact separated by few layers of water molecules separating them in a 600 mL beaker. A constant current of 0.6 A was passed through the electrochemical cell for a period ranging from two to three days. At the end of the electrolysis, many graphene quantum dots were obtained. It was stored in amber-colored bottles (see Supplementary Materials S1). An argon gas (99.99% pure) cylinder was obtained from Linde, Fulton, NY, USA, and was passed through a distilled water scrubber tube before passing into the electrochemical cell. Copper plates were obtained from Goodfellow Corporation, (Coraopolis, PA, USA).

The electrolytes for all the electrochemical experiments were 0.1 M $K_2SO_4$ in deionized water at ambient temperature with a natural pH of 6.5, with dissolved oxygen expelled by passing argon gas.

Instruments: An Ametek VersSTAT-4 Potentiostat/Galvanostat was used in all the experiments. An EDX-8100 X-ray fluorescence spectrometer was used in the measurements of elemental analysis. Electrodes: Platinum disc (A = 0.0766 cm$^2$), graphite rod (diam. 0.5 cm, Length 6 inch), and saturated calomel electrodes were used in the electrochemical measurements. The copper plate was used in the large-scale depositions. A Bruker Optik Imaging spectrometer was used for recording the Raman spectrum.

Scanning electron microscopy (SEM): The Tescan Vega-3 LMU SEM with a Bruker Xflash Detector 630M was used, which offers the widest magnification range (2.5× to 1,000,000×) in SEM, from an ultra-low magnification "macro" imaging to high-resolution imaging. The operating voltage of 20 kV was used. For XRD, a Bruker D8 was used at a current of 40 mA and voltage of 40 kV with a monochromator slit of 0.2 mm. Energy dispersive X-rays (EDXs) were used for obtaining the elemental composition.

Thermogravimetric analysis (TGA): The instrument (TGA 550) was procured from TA instruments (Delaware) and the measurements were carried out with a temperature ramp of 10 °C with oxygen gas flowing at 10 mL/min. The experiments were carried out in a temperature range of 25–1000 °C.

Electrochemical cell: Bob's cell was obtained from Gamry Instruments and was used for the measurements. Cyclic voltammetry experiments were carried out at an ambient temperature with an initial potential of the electrode set at +0.20 V with a reversal of sweep in the potential range of −0.80 V to −1.8 V. All the experiments were carried out with a glassy carbon electrode as the working electrode, a graphite plate as the counter electrode, and a saturated calomel electrode as the reference electrode.

*Deposition Conditions*

The experimental depositions of the intermetallic Cu-Zn-C was carried out on copper and graphite substrates from a medium containing 0.1 M sodium sulfate, 0.02 M copper sulfate, 0.018 M zinc sulfate, and 1 mL GQD in a potentiostatic experiment at pH 6.5, with the potentials controlled at either −0.60 V or −1.20 V. The solutions were continuously stirred by argon gas for the electrolysis duration of 1800 s. A very good adherent deposit was obtained. The deposit was washed for several hours with distilled water and air dried before examining the samples by AFM, XRF, and EDAX.

## 3. Results and Discussion

The electrodeposition of metals, especially the atomic layers, has been of great interest in technological applications. The electrodeposition of copper has been of high interest in interconnections in microelectronics [63–71]. A typical cyclic voltammetry curve of cupric ion is typified by a cathodic peak potential at $E_{pc} = -0.13$ V vs. SCE where two electron reductions occur, as shown by Equation (1) (see also Supplementary Materials S2).

$$Cu^{2+} + 2e^- = Cu \tag{1}$$

The reversal scan shows a strong anodic peak at $E_{pa} = 0.016$ V vs. SCE. By examining the cathodic peak, the current decays smoothly after the peak, as expected for the boundary conditions involved [72,73]: $C_{cu^{2+}}(x,0) = C^\circ_{cu^{2+}}$, $C_{cu^{2+}}(0,t)$ (t > 0) = 0 (past the peak potential), and $C_{cu^{2+}}(x,t) = C^\circ_{cu^{2+}}$ (x → ∞). The flux of diffusing in and diffusing out species are equal. The boundary conditions predict that the current decay past the peak potential follows a typical Cottrell's equation [72]. Indeed, the experimental conditions employed fulfilling the boundary conditions shows the expected behavior. It is further supported by the potential-step (chronocoulometry) experiments described later. In order to observe whether it would be possible to deposit zinc along with copper at the diffusion-limited transport conditions for the cupric ion to the electrode, in a situation where the zinc ion is cathodically reduced at a highly negative potential ($E_{pc} = -1.21$ V), the current was monitored at −0.60V after the addition of the zinc ion into the medium. The current value was found to be unperturbed and hence it is concluded that the zinc ion is not reduced along with the cupric ion. The electrochemically deposited copper, when examined by XRF, showed no trace of zinc in the sample. The addition of GQD to the medium showed significant changes in the current values obtained at −0.60V. Table 1 shows the measured current values under different conditions; a significantly higher flow of current when GQD was present in the electrolytic bath suggested that the zinc ion is being reduced at a positive potential than the value observed in a medium without GQD. The experiments carried out with the zinc ion in a medium with and without GQD showed peak potentials at $E_{pc} = -1.21$ V and $E_{pc} = -0.90$ V, respectively. This shift is similar to what has been reported in the literature when a complexing medium was present in the bath. Figure 1 shows the cyclic voltammetry curves when both cupric ions and zinc ions are present in the medium.

**Table 1.** The measured currents from the cyclic voltammetry.

| $C_{cu}^{2+}$ (mM) | $C_{zn}^{2+}$ (mM) | $C_{GQD}$ | I (uA) | Sweep Rate (mV/s) |
|---|---|---|---|---|
| 1.31 | 0 | 0 | 11.2 | 20 |
| 1.31 | 0.65 | 0 | 10.8 | 20 |
| 1.31 | 1.30 | 0 | 9.1 | 20 |
| 2.56 | 1.28 | 0 | 13.7 | 20 |
| 1.25 | 5.00 | 1 mL | 25.6 | 20 |
| 2.73 | 1.36 | 1 mL | 38.4 | 20 |
| 1.28 | 2.56 | 0 | 13.7 | 20 |
| 0.69 | 1.38 | 1 mL | 7.22 | 20 |
| 1.31 | 0 | 0 | 11.2 | 20 |

There are two distinct cathodic and anodic peaks due to reductions in the cupric and zinc ions. The observed features depend on the ratio of cupric ion to zinc ion in the medium. Table 2 gives the peak current ratio for the two redox couples involved in the medium. The first cathodic peak shows a sharp anodic peak, whereas the second cathodic peak that is attributed to the zinc ion reduction shows a diffuse peak. The ratio of anodic to cathodic peak currents for cupric ion is more than one whereas for the zinc ion the same ratio is less

than one. The cyclic voltammetry pattern of the zinc ion alone shows the characteristic cathodic and anodic peaks (see Supplementary Materials S3). When GQD is present in the medium along with cupric ions, the cathodic peak of the zinc is shifted towards a positive potential with no corresponding anodic peak being present. Only a single anodic peak is observed at +0.10 V. The appearance of a single anodic peak when both cupric and zinc ions are present in the medium has been discussed in the literature [74]. When ethylene diamine tetra acetate (EDTA) is present in the medium containing 0.14 M $CuSO_4$ and 0.06 M $ZnSO_4$, a well-defined single anodic peak was observed, which has been attributed to the Cu-Zn alloy in the β and γ phases by the isolation of homogeneous thin crystallites [8,75].

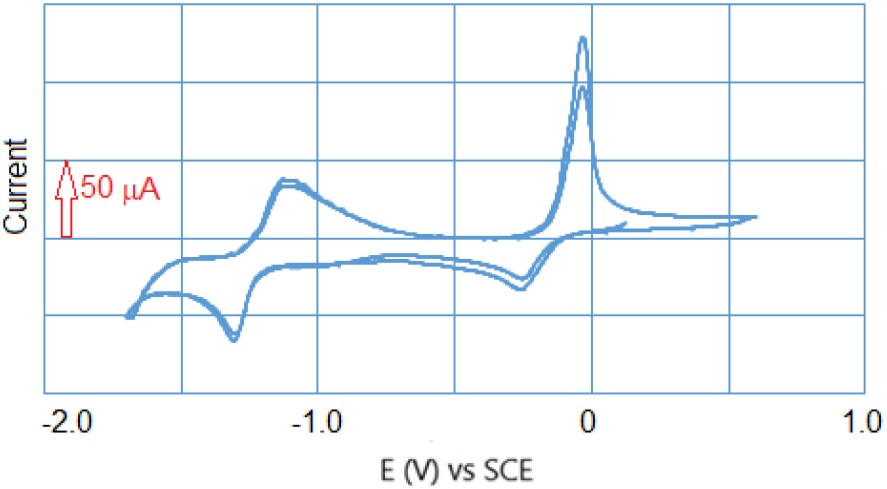

**Figure 1.** 0.97 mm $CuSO_4$ and 1.94 mm $ZnSO_4$ in a 0.1 M $K_2SO_4$ glassy carbon working electrode. No GQD in the medium.

**Table 2.** The cyclic voltammetry features of copper sulfate and zinc sulfate in 0.1 M $K_2SO_4$.

| Conc. $CuSO_4$ mm | Conc. $ZnSO_4$ mm | Sweep Rate (V/S) | $i_{pc}$ (uA) | $i_{pa}$ (uA) | $i_{pa}/i_{pc}$ |
|---|---|---|---|---|---|
| 1.10 | 0 | 0.02 | 15.8 | 60.6 | 3.83 |
| | | 0.05 | 19.0 | 81.2 | 4.27 |
| | | 0.10 | 23.8 | 98.0 | 4.12 |
| | | 0.20 | 29.7 | 105.0 | 3.53 |
| | | 0.50 | 42.0 | 142 | 3.38 |
| 1.09 | 1.10 | 0.02 | 26.9 * | 23.9 * | 0.88 |
| | | 0.05 | 37.7 * | 37.5 * | 0.99 |
| 0.97 | 2.88 | 0.02 | 63.0 * | 44.2 * | 0.70 |
| | | 0.05 | 89.2 * | 67.0 * | 0.75 |

W.E. glassy carbon. * Refers to the peak current of the zinc ion redox couple.

Nucleation Modeling: In order to understand the nucleation and growth mechanism during the formation of the alloy, the current–time curves obtained during potential-step electrolysis carried out at −0.60 V and −1.40 V were analyzed. Figure 2 shows the current-decay behavior of the solution containing the cupric ions alone and a mixture of the cupric and zinc ions in the medium. Figure 2A shows the current decay in the cupric ion solution containing GQD with the current value reaching to 4.17 μA when extended to 100 s (transient analysis is restricted only to milliseconds) (see Supplementary Materials S4). When the zinc ion is present in the solution, the current decays in a similar manner; however, the current is levelled at a higher value of 11.0 μA (Figure 2B). Apparently, the zinc ion is also

contributing to the current at a potential where the cupric ion is reduced. The enhanced currents in the presence of the zinc ion in the reduction of the cupric ion is absent when GQD was not present in the bath. The current–time transients were analyzed based on the Scharifker–Hills (SH) model for instantaneous and progressive nucleation [76], which were refined by Palomar-Pardave et al. [77] and Rodríguez-Clemente [78] (Figure 2C). With instantaneous nucleation, Figure 2D shows the current decay behavior when the potential is set at −0.60 V in a medium containing zinc and cupric ions along with GQD. In this experiment, the current decays to 23.7 µA. The current–time transients were normalized through the current maximum and compared with the theoretically simulated plots based on the models. It is interesting to compare the two nucleation processes in this model. The tm and im for the progressive nucleations are equal to $\{1.487/ANkD\}^{1/2}$ and $\{0.4615\ zFD3/4C(kAN)^{1/4}$, whereas for instantaneous nucleation they are contrastingly different; the corresponding values are tm = $\{0.399/NkD\}$ and im = $\{0.6382zFDC(kN)^{1/4}$. Figure 2D gives the current and time maxima relationships [79] for multiple nucleations. These models also provide the differences in the current maximum reached in the current–time transients for the instantaneous and progressive nucleations.

$$(i^2{}_{max}t_{max}) = 0.1629\ (zFC)^2D \quad \text{Instantaneous nucleation} \tag{2}$$

$$(i^2{}_{max}t_{max}) = 0.2598\ (zFC)^2D \quad \text{Progressive nucleation} \tag{3}$$

where $i_{max}$ is the current maximum in the current–time transient, $t_{max}$ is the time taken to reach the maximum value, z is the charge, F is the Faraday constant, C is molar concentration of the electroactive species, and D is the diffusion coefficient.

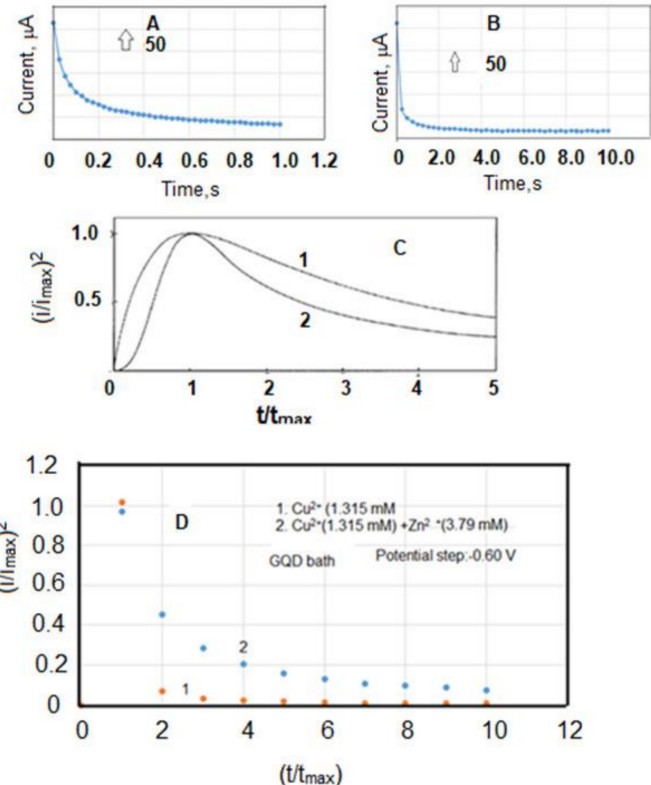

**Figure 2.** (**A**) Potential-step electrolysis at −0.60 V in a 1.31 mm cupric sulfate solution containing 1 mL GQD. (**B**) Medium: 1.31 mm $Cu^{2+}$ + 3.79 mm $Zn^{2+}$ + 1 mL GQD. (**C**) Theoretically modelled behaviors of (1) instantaneous nucleation and (2) progressive nucleation. (**D**) Nucleation model for (1) copper deposition and (2) copper deposition in the GQD bath containing zinc ions.

Notice that Equations (2) and (3) differ in the constants in front and suggests a higher value for the progressive nucleation. In the present work, the current–time transients were

used to examine the change in multiple nucleation. Table 3 shows the measured currents of the current–time transients in the potentiodynamic electrolysis. As the concentration ratio of zinc to cupric ion in the bath increased, the value of $i^2_{max}t_{max}$ decreased. This situation corresponds to the deposition of both copper and zinc in the electrolysis. Examining this behavior with the modelled predictions shown in Table 3 and Figure 2 suggests a change in the mechanism from instantaneous to progressive nucleation, as the diffusion coefficients of the copper and zinc ions are nearly the same ($D_{cu} = 5.83 \times 10^{-6}$ cm$^2$/s [79] and $D_{zn} = 1.18 \times 10^{-5}$ cm$^2$/s), and both undergo a two electron reduction during electrolysis. These results suggest that there is an interaction between graphene and the metal atoms, as predicted by Adamska et al. [80] through density functional theory calculations. Interestingly, the above behavior of change in nucleation mechanism is observed in the experiments carried out with and without zinc ions in the medium during deposition of copper in the GQD bath. Figure 2 shows the plot of $(i/i_{max})^2$ vs $(t/t_{max})$ with zinc ion present in the GQD bath; if a cupric ion reduction occurs as a progressive nucleation mechanism, the presence of the zinc ion changes it more towards instantaneous mechanism in comparison to the theoretical model described by Figure 2D. The measured currents are corrected for charging currents by using the background electrolyte electrolysis.

**Table 3.** Influence of the cupric ion in potential-step electrolysis.

| Concentration CuSO$_4$ (mM) | $i_{max}$ (A) | $t_{max}$ (s) | $i^2t_{max}$ A$^2$s | Effective ZnSO$_4$ Concentration | Ratio: (Zn$^{2+}$)/(Cu$^{2+}$) |
|---|---|---|---|---|---|
| 0 | $1.90 \times 10^{-4}$ | 0.075 | $0.27 \times 10^{-8}$ | 1.31 mm | |
| 0.65 | $2.9 \times 10^{-4}$ | 0.125 | $1.02 \times 10^{-8}$ | 1.31 mm | 2 |
| 1.29 | $5.54 \times 10^{-4}$ | 0.075 | $2.30 \times 10^{-8}$ | 1.30 mm | 1 |
| 1.93 | $6.95 \times 10^{-4}$ | 0.050 | $2.41 \times 10^{-8}$ | 1.30 mm | 0.67 |
| 2.56 | $1.49 \times 10^{-3}$ | 0.050 | $1.1 \times 10^{-7}$ | 1.30 mm | 0.50 |

### 3.1. Composite Analysis

The electrochemically deposited composite was examined for qualitative identification of the elements. Based on measurements done by SEM and Raman spectroscopy, the presence of carbon, copper, and zinc in the composite was established. As the deposit is not homogeneous within the interaction volume, having porosity with gaps, the quantitative data proved ambiguous by SEM. As the potential for deposition is in the underpotential region, the zinc content was very low (see Tables 3 and 4). XRF recordings further confirmed the presence of copper and zinc for semiquantitative analysis. XRD recordings further confirmed these results. Surface morphology has also been examined by AFM. These aspects are discussed in the following pages.

**Table 4.** Elemental analysis of the electrodeposited samples.

| Sample | C wt % | Cu wt % | Zn wt % | Substrate | Potential | Charge (C) |
|---|---|---|---|---|---|---|
| 1 | 34.01 (64.82) | 65.61 (0.11) | 0.37 (0.002) | C | −1.20 V | 34.80 |
| 2 | 86.64 (54.14) | 13.30 (0.02) | 0.05 (0.00) | Cu | −0.60 V | 19.50 |
| 3 | 97.08 (97.72) | 2.88 (0.01) | 0.03 (0.00) | C | −0.60 V | 8.91 |
| 4 | 91.03 (80.18) | 8.93 (0.02) | 0.04 (0.00) | C | −1.20 V | |
| 5 | 86.76 (53.06) | 13.00 (0.02) | 0.20 (0.00) | Cu | −1.20 V | 19.00 |

Bracket values are std values.

### 3.2. XRF Analysis

The measurement conditions for the electrodeposited samples are for C 15 kV, 982 uA-auto, acq 0–20, collimator 3 mm, analysis 0.08–0.48, tme live-100, and DT% 30; Cu 50 kV, 883 uA-auto, Filter #4, acq 0–40, analysis 7.84–8.24, live-100, and DT% 30; Zn 50 kV, 883-auto, Filter #4, acq 0–40, analysis 8.44–8.84, live-10, and DT% 30. The quantitative data obtained is given in Table 4 in helium atmosphere. The microscopic pictures of the deposits on the graphite and copper substrates are presented in Figure 3C. A survey of the literature relevant to the deposition of Cu and the Cu-Zn alloy is summarized in Table 5.

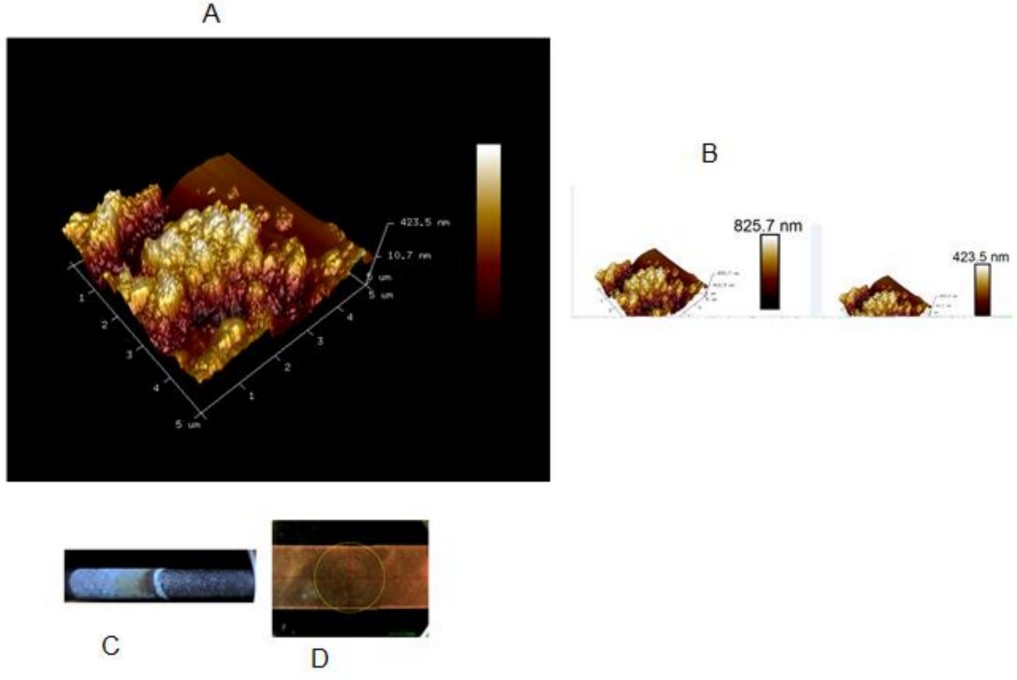

**Figure 3.** (**A**) AFM recordings of high carbon containing brass. (**B**) Particle size. (**C**) Electrodeposited high C-containing Cu-Zn alloy on a graphite substrate. (**D**) Copper substrate.

**Table 5.** Percentage weight of Cu, Zn, and C in the brass obtained from EDAX.

| Films | Potential | Medium | wt % Cu | wt % Zn | wt % C | Reference |
|-------|-----------|--------|---------|---------|--------|-----------|
| Cu | −1.1 V vs Ag/AgCl | 0.7 M $K_4P_2O_7$ + 70 mm $KH_2PO_4$ + 50 mm $CuSO_4$ | 100 | 0 | 0 | 75 |
| Cu-Zn | −1.1 V vs Ag/AgCl | 0.7 M $K_4P_2O_7$ + 70 mm $KH_2PO_4$ + 50 mm $CuSO_4$ + 50 mm $ZnSO_4$ | 100 | 0 | 0 | 75 |
| Cu-Zn | −1.4 V vs Ag/AgCl | 0.7 M $K_4P_2O_7$ + 70 mm $KH_2PO_4$ + 50 mm $CuSO_4$ + 50 mm $ZnSO_4$ | 90 | 10 | 0 | 75 |
| Cu-Zn | −0.60 V | 0.1 M $K_2SO_4$ + 1 mL GQD | 13.31 | 0.05 | 86.64 | Present work |
| Cu-Zn | −1.20 V | 0.1 M $K_2SO_4$ + 1 mL GQD | 13.00 | 0.20 | 86.76 | Present work |
| Cu-Zn | −1.10 V | 3 M NaOH + 0.25 M EDTA | 6.35 | 0.04 | 0 | 76 |
| Cu-Zn | −1.50 V | 0.1 CuSO4, 0.1 ZnO, 0.2 D-mannitol, 3 KOH | 34 | 66 | 0 | 7 |

It should be pointed out that the XRF technique has been proven to be useful for elements having atomic numbers greater than 12 (Mg). It is not a suitable for elements like carbon since the energy levels are low enough that fluorescence is absorbed by the sample. The XRF analysis is very good for metals like copper and zinc. The instrument measures the counts from copper and zinc and attributes the balance to carbon since carbon

is the only other element in the sample. It is for this reason that the standard deviation is not a true deviation arising from the direct measurement. It does not measure carbon content. The sample is a black deposit coming from the graphene quantum dots. Carbon is the inevitable element other than copper and zinc (supported by SEM/EDX and Raman spectral data). The estimate of carbon is arrived at after taking the background of the substrate.

The alloy deposition requires a complexation bath with the complex not getting incorporated into the alloy [7,8,75]. In the present work, carbon incorporation into the alloy occurs during the deposition, as revealed by XRF data (see Supplementary Materials S5). The deposition of the alloy is a function of the potential and nature of the substrate, suggesting that at −0.60 V, the moles of copper and zinc deposited are 0.212 micromoles and 0.75 nanomoles, respectively, on a copper substrate. When the potential is shifted to −1.20 V, the number of moles of zinc deposited changed to 2.9 micromoles, indicating that the deposition is undergoing under the mass transport-controlled region of the zinc ion, as indicated by the cyclic voltammetry data. On the carbon substrate, the deposition of both the metals are lower, with 0.00135 millimoles of Cu and 0.015 nanomoles of Zn at −0.60 V, which increases to 0.00679 millimoles and 0.674 nanomoles at −1.20 V. The ratio of Cu to Zn deposited changes from 90 to 10 in going from −0.60 V to −1.20 V, as a consequence of the increased amount of Zn deposition. The carbon content is a function of the amount of metals deposited on the substrate. Table 5 gives several methods reported in the literature for the deposition of the alloy, and a noticeable feature is that the metal binding complexes are released into the electrolytic bath after the metals undergo electron transfer reactions at the electrode. With the result, the alloy is deposited free of the complexes through which the metal ion is transported to the electrode. During the electrolysis, another thermodynamically feasible reaction of the zinc metal reacting with cupric ions to produce zinc ions and copper is a competing process. The deposition process is interesting in that at a potential of −0.60 V, which is a diffusion-controlled reduction region for the cupric ion-bound GQD and an under potential deposition of zinc ion-bound GQD, we observe the deposition of copper, zinc, and carbon in XRF. When the potential is shifted to −1.20 V, the deposition is continued, with a higher amount of zinc as the rate of deposition of the metals are controlled by the potential of the electrode. Hence the composition of zinc in the alloy increases when the potential is shifted from −0.60 V to −1.20 V as the deposition reaches the mass transport-controlled transport region of the zinc ion (see Supplementary Materials S2 and S3), with a competing thermodynamic exchange reaction between the deposited zinc and cupric ion in the medium.

### 3.3. Atomic Force Microscopic Features of the Deposit

The deposited samples have been analyzed for qualitative analysis of surface morphology. Figure 3 gives the observed features of one AFM image of the electrodeposited brass with a high carbon content. The surface appears to have features that are possibly clusters of deposited material. The maximum roughness (Rmax) is calculated by the difference of the highest peak height above the mean line and depth of the deepest valley below the mean line. The Rmax was 834 nm, with the distribution depicted in the figure. The roughness arithmetic average (Ra) of height deviation was 115 nm. The deposits have a compact distribution all over the substrate; a scanning electron microscopy examination further confirmed this discussion (see Supplementary Materials S6).

### 3.4. XRD Features of Brass

The electrochemical deposit of the alpha brass has been reported on steel substrate [22], which showed a peak at 2θ = 43°, corresponding to CuZn (111) with d = 2.096 A°. This deposition required the ionic liquid choline acetate. The XRD features of the laser peeling brass showed the (111), (200), (220), and (211) phases [81]; this material had an ultimate strength of 372 MPa, yield strength of 230 MPa, and with negligible fatigue fracture. The XRD pattern of the brass that has been subjected to carbon irradiation showed reflections

at 41.7°, 42.6°, 48.59°, 62.46°, 71.62°, and 78.03° due to the (320), (111), (202), (600), (312), and (102) plane reflections, respectively [82]. In the ion flux ranging 56 × 10¹² to 26 × 10¹³ ions/cm², the XRD features have not shown any significant changes. The alpha and beta brasses were examined by XRD after electro polishing [83]. The "a" brass showed distinct peaks corresponding to (111), (200), (220), (311), and (222) and "b" brass showed reflections at (110) and (220). The electrodeposited Cu-Zn in alkaline solutions with D-Mannitol has been examined by XRD for the phases and morphology of the surface [84]. The XRD of the nanobrass, $CuZn_5$, showed $2\theta$ = 37.64°, 42.22°, 43.21°, 57.69°, 68.05°, 77.82°, and 83.27° [85]. The electrochemical deposition of the composite (Figure 4) shows similar $2\theta$ features at 43.64°, 45.00°, 50.83°,65.47°, 74.75°, and 90.64°, with no reflection coming from carbon. The features observed with the present composite are nearly identical to that reported by Atay et al. [86–89] for the microstructure of brass alloys of the type $Cu_{3.8}Zn_{0.2}$ (low zinc content). Interestingly, the XRD features fit those of the face-centered copper cubic phase (JCPDS, copper file No. 04–0836 ASTM 03-1005 face-centered copper cubic phase) and zinc (JCPDS-00-004-083). The absence of carbon led us to examine in the literature whether the carbon insertion into metals are characterized by XRD. For example, in steel, where carbon is inserted into an iron atom the reflections from the different planes of the iron atom are reported with no carbon reflection visible. The new alloy does show $2\theta$ reflections due to the composite indicating that the carbon is shielded in the structure. Since the Raman imaging (see Figure 5) shows the characteristic peaks due to the D and G band along with copper, it appears that the graphene planes are inserted into the composite.

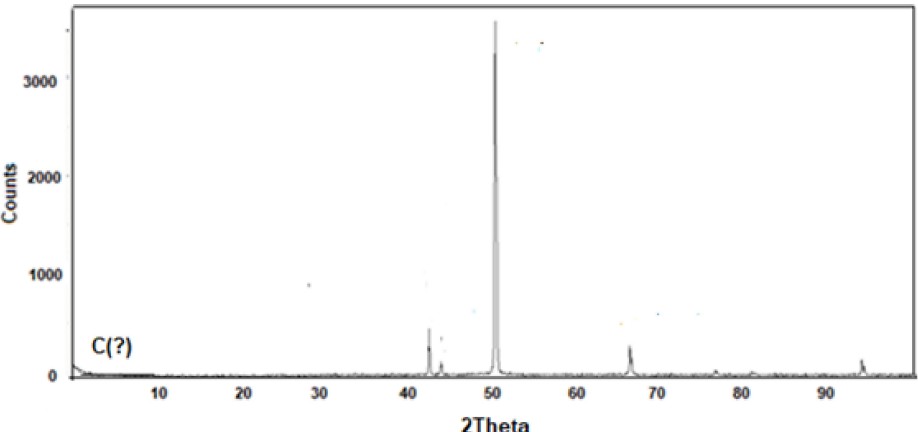

**Figure 4.** XRD of the composite obtained by electrochemical deposition in a GQD bath.

$CuK_\alpha$ = 1.5406 A°; Figure 4 exhibits 2 theta reflections at 43.64° (269.41), 45.07° (38.72), 50.83° (2966.06), 65.47° (223.97), 74.75° (33.58), and 90.64° (128.45), with the counts registered in brackets. In earlier studies (87b), the peak at the 2 theta values at 43.64°, 45.07°, and 50.83° have been assigned to CuZn(111), (110), and (200).

The powder diffraction of the cubic alpha brass with a large zinc content showed 2 theta reflections at 42.32°, 49.27°, and 72.24° (87c). Since our sample has a low zinc content due to the underpotential deposition and having a black appearance due to the GQD, the matrix may be causing the 2 theta shifts. The 2 theta reflections due to the GQD at 20° has not been observed and is attributed to the amorphous nature of the composite; similar observations have been made previously with CA/GQD composites (87d). This suggestion is further supported by the Raman spectrum of the composite where the D and G peaks have been broadened (see later section).

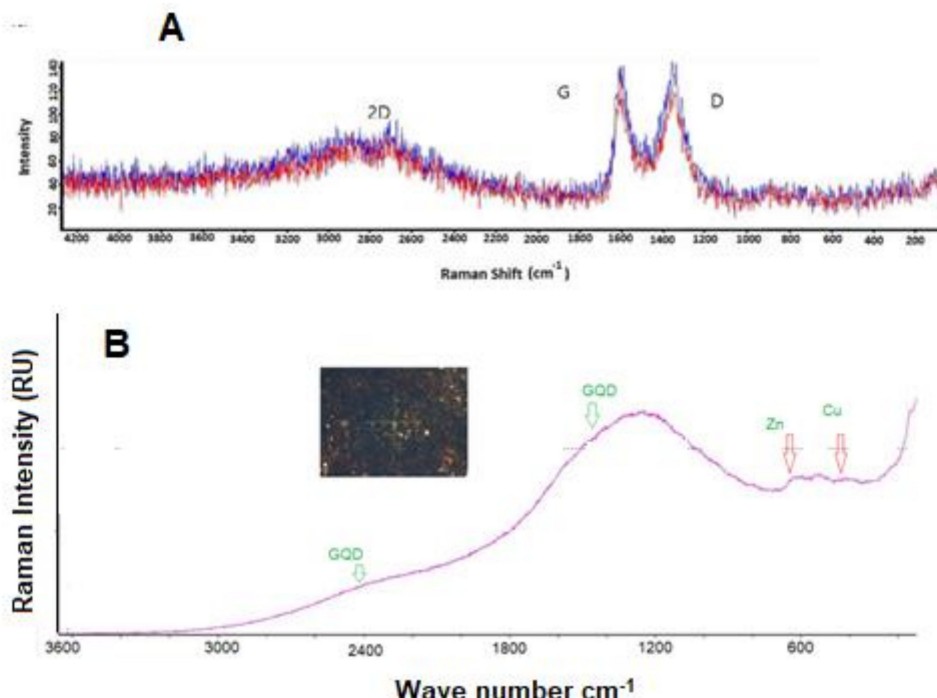

**Figure 5.** Raman spectrum of the electrochemically prepared composite GQD before (**A**) and after deposition of the composite (**B**).

### 3.5. Raman Imaging

Raman imaging spectroscopy of the electrodeposited samples from the graphene bath was examined by 632 nm laser with a variable power up to 100 mW. The image showed the presence of copper nanoparticles at 580 cm$^{-1}$ and 428 cm$^{-1}$ [90]. The broad peaks observed in Figure 5 at 1350 cm$^{-1}$ and 1450 cm$^{-1}$ are attributed to carbon in the deposited sample. All the samples examined showed an appreciable Raman intensity in the region of 3000–1400 cm$^{-1}$. We also examined the Raman imaging using 780 nm laser light; it showed distinct strong peaks at 1380 cm$^{-1}$, 1545 cm$^{-1}$, and 2690 cm$^{-1}$. The peaks are attributed to carbon in the deposited brass. There are a number of papers published in the literature on graphene Raman imaging [91–94]. The appearance of the above Raman shifts at the above wavenumbers has been attributed to sp$^2$ clusters. The broadening of the D and G bands are attributed to an electron–phonon interaction.

### 3.6. Thermogravimetric Analysis

The thermogravimetric analysis of the electrodeposited sample was done using a platinum pan. With oxygen gas flowing through the heating chamber, the new brass showed a thermal stability up to about 600 °C. At this temperature, weight gain was observed due to the oxidation of the new brass.

### 3.7. SEM of the Deposit

The scanning electron microscope recording of the alloy is presented in Figure 6 for qualitative elemental analysis. It showed distinct peaks of Cu, Zn, and C. The surface morphology showed a uniform distribution of the particles, which are crusty and mid grey in the deposits. The presence of a nodular structure of the micrograins and the absence of a lamellar structure is indicative of the contrasting behavior of the Cu$_{50}$-Zn$_{200}$ and Cu$_{100}$-Zn$_{200}$ alloys [95]. The particle distribution suggests that the high-carbon low brass would stabilize the dezincification that is observed with high-brass alloys. The EDX spectrum showed the presence of C, Cu, and Zn, which is in agreement with the XRF data discussed earlier. These results suggest that the Cu$^{2+}$ and Zn$^{2+}$ ions are being dragged to the electrode by GQD by the electric field where the electron transfer process takes place to deposit the

alloy. While conventional brass alloy is shiny metallic, the appearance of the Cu-Zn-GQD alloy is black.

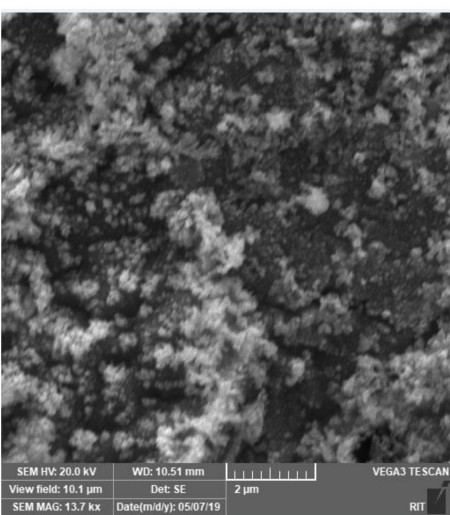

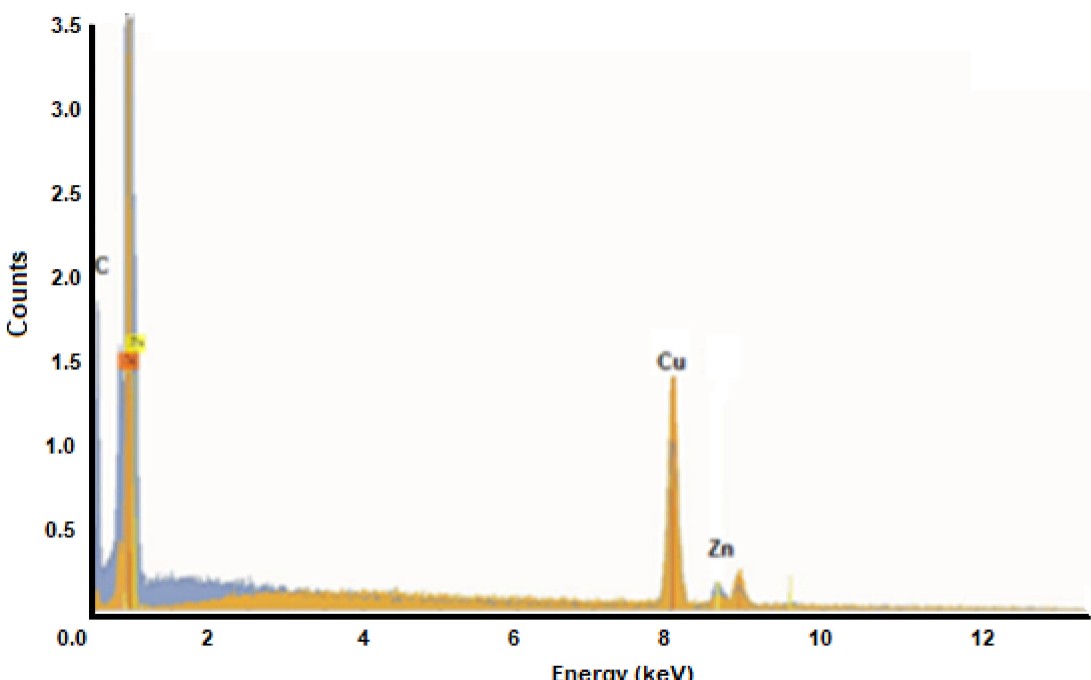

**Figure 6.** (**Upper** picture) SEM of the composite alloy; (**lower** picture) the EDX of the composite.

### 3.8. Tafel Measurements

The potentiodynamic polarization of the Cu-Zn-GQD sample was carried out in a 0.1 M NaCl electrolyte, and Figure 7 gives the Tafel plot obtained on a Cu-Zn-GQD sample obtained by carrying out the deposition on a copper plate by passing 19 C for a duration of 1800 s. From the Tafel plot a corrosion potential of $E_{corr}$ = −0.16 V and $i_{corr}$ = 0.95 $\mu A/cm^2$ were obtained. Previously [96–102] the corrosion of brass has been examined in different media with a view to reduce the corrosion by inhibitors. Kilinccer and Erbil [98] obtained corrosion rates of 405 $mA/cm^2$ by polarization studies for 67% Cu and 33% Zn in 0.1 M $Na_2SO_4$ and a reduced value of 365 $mA/cm^2$ in phosphate-containing electrolyte. Rossato et al. [99] have obtained 28.52 $mA/cm^2$ in the same electrolyte and arrived at a conclusion that oxides are being formed. The corrosion density of brass in 3.5% NaCl has

been determined to be 11.96 mA/cm$^2$ and in the presence of variamine blue B it decreased to 0.61 mA/cm$^2$ [100]. The corrosion of duplex brass has been measured in a 0.5 M H$_2$SO$_4$ medium to be 49 mA/cm$^2$, which decreased upon addition of cetyltrimethylammonium bromide to 20 mA/cm$^2$ [101]. Brass is remarkably well known for dealloying (leaching of zinc) and several approaches are being adopted for reducing the dealloying. It is reported that a small amount of P or As into brass will reduce the dezincification; the mechanism by which it retards dezincification has not been clear although it is believed that a protective layer formation occurs. The leachants, such as zinc, are often found in several waste products. Jessop and Turner [1] converted the waste products of slag by smelting through metallurgical plants. Leaching has been observed in ground samples of boat paint. At an industrial scale, the recovery of zinc and copper from the leaching of brass has been in operation in different parts of the world [2].

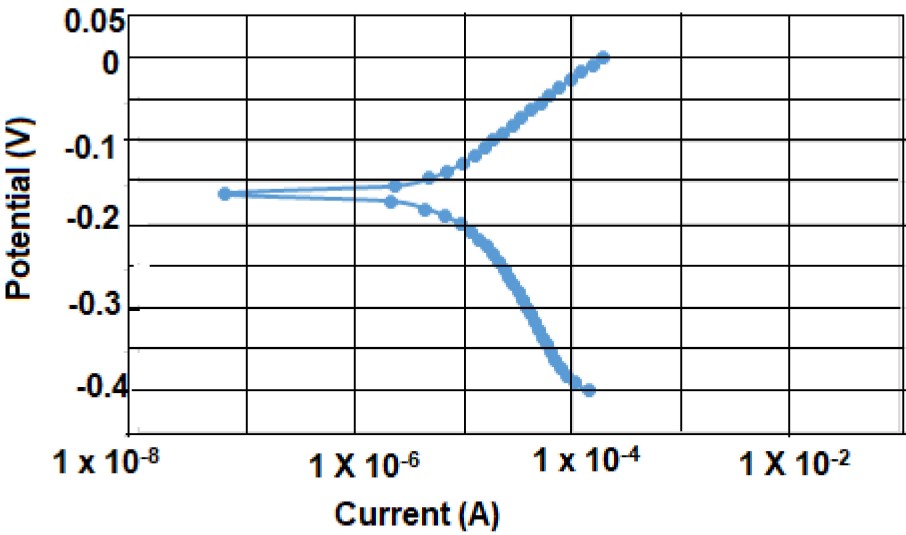

**Figure 7.** Potentiodynamic curve of the Cu-Zn GQD sample.

In the material produced in this work, Cu-Zn-GQD appears to be a revolutionary material in reducing the leaching. The E$_{corr}$ value of the Cu-Zn-GQD is less negative than the brass or that of copper [100]. Overall, the corrosion rate of this material is considerably lower than the conventional Cu-Zn alloy.

## 4. Conclusions

The electrochemical deposition of a copper composite consisting of Cu-Zn-GQD containing a high carbon content was successfully achieved using a graphene quantum dot bath. The cyclic voltammetry data with current measurements provided an insight into the mechanism. The current–time transients during potential-step electrolysis were used to model the nucleation process of the brass with a high carbon formation. X-ray fluorescence, atomic force microscopy, Raman spectral imaging, X-ray diffraction, thermogravimetry, and scanning electron microscopy with EDAX were used to analyze the composite material. The Tafel measurements were carried out on the new Cu-Zn-GQD composite alloy to determine the corrosion current density and for evaluating the sustainability of the material. The results indicate the material has a higher sustainability compared to a normal Cu-Zn alloy.

**Supplementary Materials:** The following are available online at https://www.mdpi.com/2504-477 X/5/1/9/s1; S1: Graphene quantum dots and Raman spectrum; S2: Cyclic voltammetry of copper sulfate; S3: Cyclic voltammetry of copper sulfate and zinc sulfate in the presence of GQD; S4: Excel files of current-time transients Sheet 10; S5: XRF recording of the electrodeposited composite; S6: AFM images and roughness details.

**Author Contributions:** H.R. carried out electrochemical work, including the current–time transient analysis and AFM recordings; C.B. carried out thermogravimetric analysis and Tafel experiments; B.H. carried out cyclic voltammetry and XRF recordings; R.T. carried out XRD and SEM recordings; K.S.V.S. planned the project administration, supervision, and editing. All authors have read and agreed to the published version of the manuscript.

**Funding:** The financial support for this work came from the National Science Foundation.

**Institutional Review Board Statement:** Not applicable.

**Informed Consent Statement:** Not applicable.

**Data Availability Statement:** Not available. Data is in Supplementary Materials section.

**Acknowledgments:** One of the authors (K.S.V.S.) thank the National Science Foundation for the financial support. The authors are thankful to R. Winter and T. Atkins for help with recording the XRF, AFM, SEM and XRD of the samples.

**Conflicts of Interest:** The authors declare no conflict of interest.

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
