# Peer review of "Electrodeposition from a Graphene Bath: A Sustainable Copper Composite Alloy in a Graphene Matrix"

_jcs, doi:10.3390/jcs5010009_

Round 1

Reviewer 1 Report

The author corrected the manuscript well. If the following comments are further considered, I might recommend acceptance of the manuscript as the paper which study electrochemically the GQD composite.

line 120

correct 20 KV to 20 kV

Fig. 2

Since the first data point may include double layer discharge current, the author might need to delete the first data point and re-estimate the Sharifker-Hills fitting. The key of the model is the decay curve of the (i/imax)2 after (t/tmax): if the decay curve was reproduced well by the experimental data, the electrodeposition obeys the either model.

My concern is that the use of the first data point as imax might underestimate the following (i/imax)2, especially for the data 1 in Figure2D.

Author Response

Reply to referee 1.

  1. Line 120    Thanks for the typo. It is corrected.
  2. Figure 2

In potential step electrolysis, the measured current is a total of Faradaic and charging current arising from double layer. The Faradaic current decays with square root of time, while the latter decays exponentially with time. The separation of the two currents can be realized based on the different decaying rates. In the literature, the two currents are isolated by carrying the potential step electrolysis of the background electrolyte alone with no electroactive species being present in the medium to remove the double layer charging contribution. This may be called the filtering technique. In another method, the separation of  the two currents has been accomplished by examining the two decaying rates. Here the initial target vectors based on theoretical formula of the two currents have been used to calculate the weights of them in the measured signal. The two currents in an electrochemical system have been obtained by multiplying the final target vector and the weights vector.

In the present work we have used the filtering method. The data in Figure 2D has been filtered for the double layer charging to give only the Faradaic current. We have forgotten to add this in the manuscript. We are thankful to you for bringing this to our attention. In the revised manuscript, we have added a sentence during the discussion on Figure 2D.

Reviewer 2 Report

The paper presents the qualitative evaluation of the sustainability for the produced copper composite alloy by electrodeposition from a graphene quantum dots bath.

The object of the study is worth publishing; and the paper itself presents good experimental for understanding the structure and elemental compositions.

The paper is well written and discussion can support the theory of the mechanism for the electrodeposition from graphene bath for the fabricated compositions.

Therefore, I suggest publishing this manuscript in its present form.

Author Response

Referee 2

Thanks for the comments and for the recommendation.

Round 2

Reviewer 1 Report

The manuscript has been significantly improved and now warrants publication in J. Compos. Sci.